# Development and Identification of Novel α-Glucosidase Inhibitory Peptides from Mulberry Leaves

**DOI:** 10.3390/foods12213917

**Published:** 2023-10-26

**Authors:** Fanghui Deng, Yihao Liang, Yuelei Lei, Shanbai Xiong, Jianhua Rong, Yang Hu

**Affiliations:** 1College of Food Science and Technology, Huazhong Agricultural University, Wuhan 430070, China; 2Bioactive Peptide Technology Hubei Engineering Research Center, Jingzhou 434000, China

**Keywords:** mulberry leaf peptides, α-glucosidase, molecular dynamics simulation, molecular docking

## Abstract

The mulberry leaf is a botanical resource that possesses a substantial quantity of protein. In this study, alcalase hydrolysis conditions of mulberry leaf protein were optimized using the response surface method. The results showed that the optimum conditions were as follows: substrate protein concentration was 0.5% (*w*/*v*), enzymatic hydrolysis temperature was 53.0 °C, enzymatic hydrolysis time was 4.7 h, enzyme amount was 17,800 U/g, and pH was 10.5. Then mulberry leaf peptides were separated by ultrafiltration according to molecular weight. Peptides (<3 kDa) were screened and subsequently identified using LC-MS/MS after the evaluation of α-glucosidase inhibition across various fractions. Three novel potential bioactive peptides RWPFFAFM (1101.32 Da), AAGRLPGY (803.91 Da), and VVRDFHNA (957.04 Da) with the lowest average docking energy were screened for molecular dynamics simulation to examine their binding stability with enzymes in a 37 °C simulated human environment. Finally, they were prepared by solid phase synthesis for in vitro verification. The former two peptides exhibited better IC50 values (1.299 mM and 1.319 mM, respectively). These results suggest that the α-glucosidase inhibitory peptides from mulberry leaf protein are potential functional foods or drugs for diabetes treatment, but further in vivo studies are needed to identify the bioavailability and toxicity.

## 1. Introduction

Diabetes mellitus (DM) is a prevalent non-communicable condition that is observed globally and necessitates the prolonged use of hypoglycemic medications. According to the International Diabetes Federation (IDF), there were 537 million diabetics worldwide in 2021 [1]. If no preventive measures are taken, the number will rise to 783 million in 2045 [1]. Type 2 diabetes mellitus (T2DM) is the prevailing form, constituting over 90% of the patient population. For clinical treatment, the α-glucosidase inhibitor is a type of T2DM replacement therapy that regulates postprandial blood glucose levels [2]. The process of polysaccharide degradation into glucose is impeded by the inhibition of α-glucosidase, which is situated at the brush border of the epithelial cells of the small intestine. In contrast to other medication types, α-glucosidase inhibitors possess the benefit of exerting a localized impact inside the intestinal tract, this demonstrating minimal absorption and the absence of systemic adverse effects. Recently, acarbose, miglitol, and voglibose have been used as common clinical α-glucosidase inhibitors. However, these agents still have some disadvantages, such as diarrhea, abdominal pain, and high cost [3,4]. Hence, the potential for acquiring α-glucosidase inhibitors from natural origins that are cost-effective and exhibit few adverse effects is highly encouraging.

Bioactive peptides are composed of 2–20 amino acids. These peptides have no obvious activity in the macromolecular state but show anti-diabetes, anti-hypertension, anti-inflammation, anti-cancer, and other biological activities after degradation [5,6]. Enzymatic hydrolysis is commonly employed in both research and industrial settings for the preparation of bioactive peptides due to its favorable attributes, including gentle reaction conditions and the ability to effectively regulate the reaction process [7]. Compared with traditional separation and screening technologies, the combination of molecular docking and molecular dynamics simulation are more convenient and commonly used to discover new bioactive peptides [8]. At present, α-glucosidase inhibitory peptides have been separated and identified from many natural sources. For example, wheat germ protein-derived LDLQR, AGGFR, and LDNFR [9]; egg white protein-derived RVPSLM and KLPGF [10,11]; sericin-derived SEDSSEVDIDLGNLG [12]; black bean protein-derived TTGGKGGK [13]; and spicy wood seed protein-derived KETTTIVR [14] have been reported. These bioactive peptides show excellent α-glucosidase inhibition in vitro with great application prospects. In the future, in vivo tests are necessary for the confirmation of their clinical bioability and wide application [15].

The protein of mulberry (*Morus alba*) leaf exhibits a balanced composition of essential amino acids that account for about 25% of the dry weight; thus, these proteins are cheap, abundant, and of high quality [16]. Mulberry leaves possess a diverse range of bioactivities that contribute to the enhancement of human health, including but not limited to anti-diabetes, anti-oxidation, and anti-atherosclerosis effects [17,18,19]. Although the precise chemical responsible for this benefit is unknown, the ancient Chinese medicinal book *Qi Min Yao Shu* referenced the effect of mulberry leaf as traditional Chinese medicine to alleviate diabetes. Some previous studies reported that the mulberry leaf extracts showed excellent α-glucosidase inhibitory activity [18,20]. Pre-experiment results indicated that α-glucosidase inhibitory activity of mulberry leaf peptides from our cultivated high-protein variety was closely related to the molecular weight and peptide sequence. However, the functional molecular weight distribution and key sequence of the peptides were not clear and require further research.

To develop novel bioactive peptides, the enzymatic hydrolysis of mulberry leaf protein by alcalase was optimized using the response surface method (RSM) in this study. Enzymatic hydrolysates with a high degree of hydrolysis (DH) and significantly increased α-glucosidase inhibitory activity were obtained. Using ultrafiltration and liquid chromatography with tandem mass spectrometry (LC-MS/MS), the bioactive peptides with potential α-glucosidase inhibitory activity were isolated from mulberry leaf protein and identified. Using molecular docking and molecular dynamics simulation, the inhibitory mechanism of the screened peptides was examined. Finally, a test was performed to confirm the in vitro bioactivity of the chosen peptides.

## 2. Materials and Methods

### 2.1. Materials

Mulberry leaves were provided by the Hubei Academy of Agricultural Sciences. α-Glucosidase (100 UN) was purchased from Sigma-Aldrich Co. Ltd. (St. Louis, MI, USA). Alcalase (200 U/mg) and phosphate buffer solution (PBS, pH = 7.6) were purchased from Shanghai Yuanye Bio-technology Co., Ltd. (Shanghai, China). Acarbose and 4-nitrophenyl β-D-glucopyranoside (pNPG, 99% purity) were obtained from Shanghai Macklin Biochemical Technology Co., Ltd. (Shanghai, China). Other chemicals and reagents are analytical grade and commercially available.

### 2.2. Preparation of Mulberry Leaf Protein

The mulberry leaf protein was prepared according to the method of Ning et al. with slight modification [21]. After being washed, the freshly harvested mulberry leaves were subjected to a drying process and then crushed. The mulberry leaf powder was then collected by passing it through an 80 mesh sieve. The prepared powder was extracted with 5 g/L NaOH solution, treated with 40 Hz ultrasonic wave at room temperature for 10 min, and incubated in a 40 °C water bath for 1 h. After removing the residues, the pH of the crude protein solution was adjusted to 3.0 with 1 mol/L HCl. After 0.5 h, the solution underwent centrifugation at 4000 r/min, resulting in the collection of the crude protein precipitation. To remove the small molecular impurities, the protein precipitation was placed in 0.2 mol/L PBS (pH = 7.6), dialyzed with a 1 kDa dialysis bag for 48 h, and then placed in distilled water for 24 h. The treated samples were collected and used as mulberry leaf protein after vacuum freeze-drying.

### 2.3. Preparation of Mulberry Leaf Peptides

#### 2.3.1. Single-Factor Test

Mulberry leaf protein was dissolved in distilled water before heating at 95 °C for 10 min to inactive endogenous proteases. The parameters for the enzymatic hydrolysis process were as described below. After enzymatic hydrolysis, the hydrolysates were also subjected to the same heat treatment at 95 °C for 10 min to terminate the reaction. Then, centrifugation with 8000× *g* for 10 min was applied to gather supernatants for freeze-drying.

In our previous study, alcalase was characterized as having high peptide yield and catalytic activity in enzymatic hydrolysis. Therefore, alcalase was used in this research. The effects of substrate protein solution concentration (0.5%, 1.0%, 1.5%, 2.0%, 2.5%, 3.0%, *w*/*v*%), enzyme amount (5000 U/g, 10,000 U/g, 15,000 U/g, 20,000 U/g, 25,000 U/g, E/S), enzymatic hydrolysis temperature (25 °C, 35 °C, 45 °C, 55 °C, 65 °C), pH (9.0, 9.5, 10.0, 10.5, 11.0, 11.5), and enzymatic hydrolysis time (2 h, 3 h, 4 h, 5 h, 6 h, 7 h) on the degree of hydrolysis (DH) and yield of soluble peptides (YSP) were investigated.

#### 2.3.2. Optimization of Enzymatic Hydrolysis Parameters via Response Surface Method (RSM)

According to the results of single-factor experiments and the principle of Box-Behnken design, enzymatic hydrolysis temperature, enzymatic hydrolysis time, and enzyme amount were selected to define the optimal conditions to maximize the DH as the response value. The design levels of parameters are shown in Appendix A.

### 2.4. Degree of Hydrolysis (DH) Determination

The DH of enzymatic hydrolysis was determined using the pH-stat method [22]. The following formula was used:(1)$$DH=\frac{B\times N_{b}}{\alpha \times M_{p}\times H_{tot}}\times 100,$$
where B is the consumption of NaOH (L), N_b_ is the concentration of NaOH (mol/L), α is the average dissociation degree of the α-NH group (1/α = 1), M_p_ is the total amount of substrate protein (g), and H_tot_ is the number of peptide bonds (mmol/g) in substrate protein (H_tot_ of mulberry leaf protein was 8.16 mmol/g).

### 2.5. Yield of Soluble Peptides (YSP) Determination

The YSP was determined using the Biuret method [23]. The sample solution of 1 mL was evenly mixed with 3 mL of 15% (*w*/*v*) trichloroacetic acid. After centrifugation at 4000× *g* for 20 min, 4.0 mL Biuret reagent was added to 1.0 mL of the sample. The OD value was determined at 540 nm and calculated using this formula:(2)$$\mathrm{YSP}=\frac{T_{c}}{T_{p}}\;\times \;100,$$
where T_c_ (mg/mL) is the concentration of polypeptide in the sample on the standard curve of bovine serum albumin (y = 0.0502x + 0.0997, R^2^ = 0.9997), and T_p_ (mg/mL) is the concentration of protein in the substrate.

### 2.6. Molecular Weight Distribution Determination

The molecular weight distribution was determined using 18-angle laser light scatter with a qualitative detector and a gel permeation chromatography column WTC-010S5 (Wyatt Technology Co. Ltd., Santa Barbara, CA, USA). The sample was fully dissolved in the mobile phase (40% acetonitrile) at 2 mg/mL and passed through a 0.45 μm filter membrane before injection. The operating parameters were set as follows: column temperature 25 °C and flow rate 0.3 mL/min.

### 2.7. Determination of α-Glucosidase Inhibitory Activity

Referring to the method of Fang et al. [24], 25 μL α-glucosidase solution (0.2 U/mL, PBS, pH = 7.0) was added to 25 μL of samples with gradient concentrations. After incubation at 37.0 °C for 20 min, 25 μL 4-nitrophenyl β-D-glucopyranoside (pNPG) solution (2.0 mmol/L) as substrate was added to start the reaction. The total volume of the reaction system was increased up to 100 μL by PBS. The sample blank group used PBS instead of α-glucosidase solution, and the control group used PBS instead of the sample solution. The reaction mixture was incubated at 37.0 °C for 30 min, and 100 μL Na_2_CO_3_ (0.2 mol/L) was added to quench the reaction. The absorbance value was measured and recorded at 405 nm using a microplate reader. The following formula was used:(3)$$\alpha -\mathrm{glucosidase}\;\mathrm{inhibition}\;\mathrm{rate}\;(\%)=\left(1-\frac{A_{a}-{\;A}_{b}}{A_{c}-{\;A}_{d}}\right)\;\times \;100,$$
where *A_a_* is the absorbance value of the sample, *A_b_* is the absorbance value of the sample blank group, *A_c_* is the absorbance value of the blank group, and *A_d_* is the absorbance value of the blank control group.

### 2.8. Crude Separation of α-Glucosidase Inhibitory Peptides

Referring to the method of Liu et al. with sight modification [9], the mulberry leaf peptides were graded using an ultrafiltration stirring cup (Merck Millipore Co. Ltd., Burlington, VT, USA) equipped with 3 kDa and 10 kDa ultrafiltration membranes (Merck Millipore Co. Ltd., USA). They were divided into three fractions: fraction 1 (<3 kDa), fraction 2 (3 kDa–10 kDa), and fraction 3 (>10 kDa). Samples were collected and freeze-dried using a vacuum freeze-dryer for further treatment.

### 2.9. Peptide Sequence Identification

Samples were subject to 12,000× *g* centrifugation for 10 min. Then, the supernatant was filtered with 10 kDa ultrafiltration tubes, and the filtrate was desalted using a C18 column. The peptide solution was created after treatment using a centrifugal concentrator. The EASY-nLC 1200 system (Thermo Fisher Scientific, Waltham, MA, USA) coupled with a C18 analytical column (75 μm × 25 cm, 2 μm, 100 Å) and Q Exactive HF-X mass spectrometer (Thermo Fisher Scientific, USA) was used for peptides sequence analysis. The mobile phase A (0.1% formic acid) and the mobile phase B (0.1% formic acid, 80% acetonitrile) were used to establish a 100-min analysis gradient. The flow rate was 300 nL/min. The data were collected in DDA mode, and every scan cycle included a full MS scan (R = 60 K, AGC = 3 × 10^6^, max IT = 20 ms, scan range = 350–1800 *m*/*z*), followed by 25 MS/MS scans (R = 15 K, AGC = 2 × 10^5^, max IT = 50 ms). HCD collision energy was set to 28. The filter window of the quadrupole was set to 1.6 Da. The dynamic elimination time of ion repeated collection was set to 35 s. Data were analyzed using MaxQuant 1.6.6 (https://www.maxquant.org/ (accessed on 1 July 2022)). The UniProt database (http://www.uniprot.org/ (accessed on 1 July 2022)) search was performed as follows. The protein variable modifications were oxidation (M). The MS match tolerance was set at 20 ppm for the initial search and 4.5 ppm for the primary search. The MS/MS match tolerance was also set at 20 ppm.

### 2.10. Virtual Screening and Molecular Docking

All identified peptides were constructed using AlphaFold2. The crystal structure of α-glucosidase (PDB ID: 5ZCE, resolution 1.55 Å) was obtained from the PDB protein database (https://www.rcsb.org/structure/5ZCE (accessed on 1 August 2022)). The enzyme underwent dehydration using PyMOL 2.5.5 (education version, Schrodinger LLC, New York, NY, USA) and subsequently treated using MglTools 1.5.6 (The Scripps Research Institute, San Diego, CA, USA) for removal of heteroatoms, the addition of missing hydrogen atoms, and merging of lone pair electrons. AutoDock vina 1.2.3 (Scripps Research, La Jolla, CA, USA) was employed for molecular docking. The center point of the docking box was X = 3.234, Y = 50.597, and Z = 80.37, and the dimensions were x = 92.75, y = 69.19, and z = 92.75. The inhibition of α-glucosidase of peptides was predicted by the free energy of docking binding. PyMOL was used to process and visualize the results.

### 2.11. Molecular Dynamics Simulation

The stability of the complex was measured using GROMACS 5.1.5 (SciLife Lab., Solna, Sweden) and the Amber99sb-ILDN force field [7]. The complex was enclosed in a cubic periodic box with a 1 nm gap between the protein and the box’s edge. The TIP3P water model was added to the box, and the calculated charge balance function was used to automatically replace the water molecules with Na^+^ to maintain the charge balance. The steepest descent method was employed to perform energy minimization and stopped when the maximum force was less than 10 kJ/mol. In this process, the minimum energy paths were 50,000, and the minimum energy step size was 0.01. The Particle Mesh Ewald method was used to evaluate the long-range electrostatic interaction. Non-bond interactions were updated at every step (nstlist = 1). The leapfrog integral method was used, and the integration step was specified as 2 fs. The molecular dynamics simulation with 50 million steps was carried out for each 100 ns analysis system at 310 K (simulated human ambient temperature). The changes in root mean square deviation (RMSD), root mean square fluctuation (RMSF), the number of hydrogen bonds between ligands and receptors, radius of gyration (R_g_), and solvent-accessible surface area (SASA) were analyzed.

### 2.12. Peptide Synthesis

Peptides (purity > 98%) were synthesized by GL Biochem, Ltd. (Shanghai, China) using the Fmoc solid-phase peptide synthesis method. The bioability of synthesized peptides was verified by the method described in Section 2.7.

### 2.13. Statistics Analysis

The data were processed and analyzed using SPSS Statistics V25.0 (IBM, Armonk, NY, USA), Origin 2021b (OriginLab Co. LTD, Northampton, MA, USA), and Prism 9 (GraphPad Software, San Diego, USA). The difference was accepted as significant at *p* < 0.05.

## 3. Results and Discussion

### 3.1. Optimization of Enzymatic Hydrolysis

#### 3.1.1. Results of Single-Factor Experiments

The degree of hydrolysis (DH) represented the percentage of free amino nitrogen content in the total amino nitrogen content in the proteolytic solution, which directly indicates the degree of peptide fragmentation. The yield of soluble peptides (YSP) indicated the ratio of protein content in hydrolysates to total protein in the substrate, which indicates the utilization degree of protein in enzymatic hydrolysis. The impact of substance concentration, temperature, enzyme amount, pH, and hydrolysis time on the hydrolysis of mulberry leaf crude protein (protein content: 746.60 ± 18.52 g/kg) was evaluated based on these two indexes.

As shown in Figure 1a, the DH significantly decreased from 21.25% to 15.8% as the substrate concentration increased from 0.5% to 2.0%. The DH was maintained at about 16% when the substrate concentration was 2.0–3.0%. This finding may be due to the increasing substrate protein concentration affecting the fluidity of the enzymatic hydrolysis system and the size of the contact surface between protease and substance [25]. The YSP of every group fluctuated around 60% within 5%. Therefore, the substance concentration of 0.5% was used for further optimization. Contrarily, there was a increasing trend in DH with the increasing temperature (Figure 1b). The value reached the peak of 23.52% at 55 °C and sharply dropped to 19.8%. Temperature can mediate the thermal stability of alcalase and then affect the process of enzymatic hydrolysis. The value of YSP did not change significantly from 35 °C to 65 °C. Accordingly, 55 °C was considered the optimal temperature. As noted in Figure 1c, the DH showed an overall upward trend with elevated enzyme amount and stopped significantly increasing after 15,000 U/g, which may be due to the presence of insufficient substrates to react with excess enzyme. The YSP exhibited the highest value of almost 70% at 20,000 U/g. Therefore, no relationship was noted between DH and YSP. Regarding pH, the YSP (57.1%) was the highest at pH 10.0, and there was no significant difference between other groups (Figure 1d). The DH value suddenly increased to reach almost 24% at pH 10.5 and subsequently plateaued. Figure 1e indicated that the DH first increased and then tended to be flat with the changes in hydrolysis time. The DH reached the maximum value at 7 h, but this value did not significantly differ compared with the value obtained at 6 h. Simultaneously, YSP reached the maximum of 77.0% and decreased rapidly at 7 h. With the increasing hydrolysis time, the substrate became limited, and the products influenced the pH of the system, making it unsuitable for enzyme function.

#### 3.1.2. Results of Enzymatic Hydrolysis Optimization

According to the results of the single-factor experiments, the response surface method (RSM) was carried out at a fixed pH of 10.5 and substrate concentration of 0.5%. The reaction temperature (A), reaction time (B), and enzyme amount (C) were selected, and DH (R) was taken as the response value to analyze the effect. Using a multiple regression fit for the data of Appendix A, the following predictive equation was obtained:(4)$$R=6.56-0.86A+1.05B+1.00C-0.83AB-0.080AC-0.15BC-2.89A^{2}-1.31B^{2}-1.02C^{2}$$

ANOVA results are shown in Table 1. The *p*-value of the model was less than 0.0001, and the *p*-value of the misfit term was 0.2458, which indicated that the influence of unknown factors on the experimental results was not significant. Thus, the fitted equation was credible. The experimental data fit well with the regression mathematical model (R^2^ = 0.9902), which can better predict the actual values of each parameter. The degree of the effect of parameters on the R-value can be judged using the F-value. In comparison to the chosen indexes, it was seen that the hydrolysis time exerted the most significant effect on the DH followed by the enzyme amount and temperature.

Based on the predicted equation, the response surface plots and contour maps were obtained (Figure 2). The shape of the contour and the steepness of the response surface represented the effects of parameters on the DH. The contour maps presented in Figure 2a,b exhibited an oval shape, indicating significant interactions between temperature and hydrolysis time or enzyme amount. In contrast, a circle-shaped graph is presented in Figure 2c, representing an insignificant interaction. Moreover, the steepness of the response surface along the time direction was greater in Figure 2a, and that along the enzyme amount direction was greater in Figure 2b. That is to say, the enzyme amount and enzymatic hydrolysis reaction time had greater effects on the DH than temperature, which was consistent with the results of Table 1.

The optimal conditions for enzymatic hydrolysis were determined to be a temperature of 53.40 °C, a hydrolysis time of 4.67 h, and an enzyme amount of 17,799.72 U/g, as shown by the highest point on the 3D surface. Given the current experimental conditions, the factors were set as follows: the temperature was 53.0 °C, the time was 4.7 h, and the enzyme amount was determined to be 17,800 U/g. The measured DH value was found to be 26.53 ± 0.21%, which closely approximated the anticipated value of 26.93%. In a comprehensive manner, the model demonstrated a good match. Moreover, a previous study on hemp (*Cannabis sativa*) seed peptides with α-glucosidase inhibitory activity hydrolyzed by alcalase revealed that the inhibition effect of the α-glucosidase strengthened with the increasing DH and weakened after reaching the maximum DH value of 27.5% [26]. The optimized hydrolysis products with a similar DH value potentially demonstrate effective inhibition as determined in the following experiments.

### 3.2. Evaluation, Separation, and Identification of α-Glucosidase Inhibitory Peptides

#### 3.2.1. Evaluation of Enzymatic Hydrolysis Products

In this work, the peptides produced by enzymatic hydrolysis with a DH of 15% had many macromolecular proteins and promoted the function of α-glucosidase (Figure 3a). Similar results in another study demonstrated that peptides obtained by alcalase at a low DH served as α-glucosidase activators [26]. It could be interpreted that the structure of some crude products from mulberry leaf protein produced under low DH conditions is similar to that of carbohydrate hydrolases. Alternatively, the products potentially synergized with α-glucosidase, which promoted the decomposition of p-nitrophenyl-β-D-glucopyranoside (pNPG) to p-nitrophenol (pNP) followed by the increased OD value at 405 nm. The molecular mechanism of this phenomenon needs to be further studied.

As shown in Figure 3c, the small molecular weight group accounted for a higher proportion of the mulberry leaf peptides obtained using enzymatic hydrolysis with a DH of 26.53% after optimization due to a higher degree of protein molecular chain breakage. Specifically, the molecular weight was 4.25% between 10 and 20 kDa, 2.04% between 5 and 10 kDa and 59.89% between 0 and 1 kDa. Compared with the before group with a DH of only 15%, their percentages increased or decreased by 10.76%, 23.07%, and 25.90%, respectively. Generally, a high DH is usually indicative of more low molecular weight peptides, resulting in a more effective inhibitory effect [27,28]. The optimized peptides showed inhibition of α-glucosidase as demonstrated in Figure 3b. However, its IC50 was 27.33 mg/mL, which is considerably greater than that of acarbose (0.43 mg/mL). There was a wide molecule weight distribution of optimized products with an average molecular weight of 2.7 kDa. Therefore, it was necessary to further isolate and identify the more effective fragment.

#### 3.2.2. Separation and Identification of α-Glucosidase Inhibitory Peptides

Three fractions (fraction 1: <3 kDa, fraction 2: 3 kDa–10 kDa, and fraction 3: >10 kDa) of enzymatic hydrolysates were obtained by ultrafiltration. Since fraction 3 still contained protein rather than peptide, the inhibitory activities of the other two fractions were detected. Figure 3d shows that the IC50 of fraction 1 and fraction 2 were 5.27 mg/mL and 14.27 mg/mL, respectively. The inhibitory effect of fraction 1 was significantly better than that of fraction 2. This finding was consistent with the results of previous studies on bioactive peptides given that the fragments with lower molecular weight exhibited higher activity [25,29,30,31]. Small molecular peptides were more likely to enter the active pocket of protein receptors and combine with their active sites. Thus, fraction 1 was considered to be an effective source of α-glucosidase inhibitor for further exploration. In order to perform the virtual screening for α-glucosidase inhibitory peptides, LC-MS/MS was used to identify the peptide sequence of fraction 1. There were 65 peptides identified with molecular weights ranging from 769.38 Da to 1283.71 Da as shown in Appendix A. Four of these were decapeptides, two were nonapeptides, and the remaining were octapeptides. Many novel bioactive peptides have amino acid numbers in this range. Briefly, identified peptides from fraction 1 were reliable sources of α-glucosidase inhibitory peptides. Without additional screening, the binding energy analysis of 65 peptides with α-glucosidase inhibitory activity was performed to discover more potential α-glucosidase inhibitory peptides.

### 3.3. Virtual Screening and Molecular Docking

#### 3.3.1. Binding Energy Analysis

Molecular docking has emerged as a bioinformatics-based approach for assessing the interaction between receptors and ligands, hence proving to be an effective technique for the screening of bioactive peptides [8]. The binding modes and affinities of enzymes and peptides are predicted using the binding energy. The peptides identified from Fraction 1 were built using Alphafold2, and the inhibition effect was analyzed by molecular docking. Each peptide was docked 10 times to obtain the results shown in Appendix A. The lower energy corresponded to a stronger binding effect. The binding energy of 650 times docking varied from −8.81 kcal/mol to −5.77 kcal/mol. Interestingly, the value in each docking was less than −6.0 kcal/mol except for EEEDKKVE and TASKLLLR. It is generally believed that a binding energy of docking less than −6.0 kcal/mol indicates effective binding [32]. The effective binding of most peptides to enzymes may be the reason for the high inhibitory activity of fraction 1. The top 10 peptides that demonstrated the most pronounced binding effects are VLPAHKFG, RRYVRQLP, SDVYAPRS, VFPKQHIF, SALPVGIW, AAGRLPGY, VVRDFHNA, RWPFFAFM, VGINCAPP, and LFYRRARK.

After calculating their average binding free energies, the comparisons between peptide-receptor and acarbose-receptor are illustrated in Figure 4. Among the ten peptides, AAGRLPGY, VVRDFHNA, and RWPFFAFM with the lowest average binding energies were considered potential inhibitory peptides. In particular, the average docking energy of RWPFFAFM (−8.65 kcal/mol) was similar to that of acarbose (−8.84 kcal/mol). Many studies concluded that hydrophobic amino acids at the C-terminal of peptides, especially leucine and proline, played an important role in the α-glucosidase inhibition of bioactive peptides and may be related to the formation of hydrogen bonds [30,33,34]. In addition, Ibrahim et al. declared that the peptides with alanine at the C-terminal residue had a lower binding free energy than the peptides with methionine at the C-terminal residue [35]. The opposite result appeared in this study; namely, the binding energy of RWPFFAFM (methionine at the C-terminal) was lower than that of VVRDFHNA (alanine at the C-terminal). This may be due to another factor affecting the inhibitory activity or the content of hydrophobic amino acids in the peptide [36]. More hydrophobic amino acids are present in RWPFFAFM (87.5%) than in VVRDFHNA (50%). Overall, these 3 peptides were identified as potential α-glucosidase inhibitors. Therefore, they were deeply analyzed and their interactions with α-glucosidase were explored.

#### 3.3.2. Binding Site Analysis

The docking results with the lowest binding energy of selected peptides are presented in Figure 5. All three peptides exhibited similar binding to acarbose in the active pocket of the enzyme. The hydrophobic interaction, salt bridge, and hydrogen bond maintained the stability of the peptide-α-glucosidase (PDB ID: 5ZCE) complex [36]. Among them, hydrogen bonds played a primary role in the inhibition of bioactive peptides, and the number of hydrogen bonds was essential [27].

VVRDFHNA-5ZCE formed eight hydrogen bonds at GLU-141, SER-145, ASP-199, GLN-256, ASP-327, GLN-328, TYR-388, and THR-409 (Figure 5b). All hydrogen bonds in this complex with a length less than 3.5 Å were considered to make a close contact, and the binding energy obtained was −8.78 kcal/mol [9]. Figure 5c demonstrates that RWPFFAFM made three hydrogen bonds with the receptor protein at ASP-60, ASP-327, and GLN-392 with lengths of 2.2 Å, 2.4 Å, and 3.5 Å, respectively. Along with other interaction forces, the binding energy was −8.75 kcal/mol. In addition, AAGRLPGY formed seven hydrogen bonds at GLU-141, GLN-167, HIS-203, ASP-382, and GLN-392 in the protein as evidenced in Figure 5d. Among them, GLU-141 and ASP-382 both formed two hydrogen bonds, and the average length of these four hydrogen bonds was 2.2 Å, which contributed to the binding free energy of −8.77 kcal/mol. As an effective α-glucosidase inhibitor, acarbose was bound to many amino acid residues of the enzyme as shown in Figure 5a. The selected peptides in this study had the same active binding sites (ASP-60, GLN-167, ASP-199, HIS-203, GLN-256, ASP-327, and GLN-328) as acarbose, revealing their binding efficacy. As noted in previous studies, chalcone compounds, astilbin, morin, and trilobatin had the same binding sites with α-glucosidase at ASP-60, ASP-199, HIS-203, GLN-256, and GLN-328, which further confirmed the bioactive potential of the screened peptides [24,32,37].

Thus, analysis of hydrogen bond binding sites revealed that effective binding between peptides and enzymes and provided a basis for molecular dynamics simulation.

### 3.4. Molecular Dynamics Simulation

Molecular docking is used to elucidate the structure–activity relationship between enzymes and bioactive peptides [20]. However, assessing the dynamic binding information solely based on static molecular docking is unfeasible. Molecular dynamics simulation in the simulated human environment at 37 °C was applied in this study to verify the results of docking and further explain the dynamic binding effect [38].

#### 3.4.1. Root Mean Square Deviation Analysis

Root mean square deviation (RMSD) represents the average displacement deviation in a specific simulation duration of the complex to reference, and a lower value indicates better stability of the system [39]. As demonstrated in Figure 6, the receptor protein in every docking system was relatively stable with an RMSD value less than 0.25 nm. Among these ligands, acarbose and VVRDFHNA had minimal movement, and their RMSF values were stable after 50 ns. The RMSD values of AAGRLPGY and RWPFFAFM fluctuated at 0.3 nm and 0.4 nm, respectively, during the 100 ns simulation. The high value of RWPFFAFM may be caused by its high content of hydrophobic amino acids. However, the ligands were not separated from the active pocket of α-glucosidase in the process of simulation, indicating the stability of complexes.

#### 3.4.2. Root Mean Square Fluctuation Analysis

As displayed in Figure 7a, root mean square fluctuation (RMSF) referred to each residue shift of α-glucosidase in four docking systems [40]. The α-glucosidase-acarbose complex with an average RMSF value of 0.0996 nm had more stable residues than α-glucosidase-VVRDFHNA, α-glucosidase-AAGRLPGY and α-glucosidase-RWPFFAFM (with RMSF values of 0.168 nm, 0.184 nm, and 0.201 nm, respectively). However, all docking systems had a similar fluctuation trend. At amino acid residues 290–300, there were irregular structures that caused the abrupt fluctuations. Meanwhile, the essential residues of the α-glucosidase binding pocket were stable, such as ASP-60, HIS-203, and ASP-327, and the RMSF value was less than 0.10 nm. Thus, the systems inside the simulated human environment were characterized by stability and cohesive connections despite the occurrence of some dynamic shifts.

#### 3.4.3. Radius of Gyration Analysis

Figure 7b presents the radius of gyration (Rg) of complexes, with a smaller value indicating a tighter structure [41]. All docking systems became stable after 50 ns. The compactness of structures from strong to weak exhibited the following order α-glucosidase-acarbose, α-glucosidase-VVRDFHNA, α-glucosidase-AAGRLPGY and α-glucosidase-RWPFFAFM. Their average Rg values were 2.37 nm, 2.41 nm, 2.42 nm, and 2.45 nm, respectively. These values were similar to the value of 2.40 nm reported in a previous study on α-glucosidase peptides [42]. Among these peptides, the densest system was the VVRDFHNA complex with high conformational stability, which was consistent with the results of RMSD and RMSF analysis presented above.

#### 3.4.4. Analysis of the Number of Hydrogen Bonds 

The hydrogen bond is an important non-covalent binding force for stabilizing the ligand–receptor complex. The change in hydrogen bonds in the docking systems is shown in Figure 7c. Acarbose, AAGRLPGY, VVRDFHNA, and RWPFFAFM formed an average of 11.09, 4.00, 8.80, and 6.83 hydrogen bonds with α-glucosidase, respectively. It seemed that VVRDFHNA and RWPFFAFM exhibited more stable binding with enzymes at dynamically simulated human temperatures. In addition, the number of hydrogen bonds in RWPFFAFM-protein complex exhibited the smallest change, which may be related to the length of hydrogen bonds formed.

#### 3.4.5. Solvent Accessible Surface Area Analysis

The solvent accessible surface area (SASA) of different complexes reveals the enzyme folding stability (Figure 7d) [20]. The SASA values of all complexes tended to be stable and had similar changes within 10 nm^2^. Their fluctuations were significantly lower than that (about 20–30 nm^2^) of pentapeptide-enzyme complexes in another study [42]. The values of peptide groups were mainly concentrated at 225–230 nm^2^, which was significantly lower than that of the acarbose group. It could be interpreted that smaller molecules had a smaller impact on the interaction between receptor protein surface and water. Generally, complexes did not exhibited dramatic changes because of the solvents in the simulation.

The binding state of the three peptides to the receptor protein was stable at simulated human temperature in the 100 ns process. The relative position of complexes was not strongly shifted with no disintegration, and stable bonds were located at the active pocket of the receptor protein.

### 3.5. Verification of the Inhibitory Bioability of Selected Peptides

To further verify the synthesis of peptides, in vitro activity tests were carried out. Their inhibitory rate curves are displayed in Figure 8. The IC50 values of AAGRLPGY, VVRDFHNA, and RWPFFAFM were 1.319 mM (R^2^ = 0.94), 2.123 mM (R^2^ = 0.90), and 1.299 mM (R^2^ = 0.96), respectively. Given the lack of a clinically effective peptide that functions as an α-glucosidase inhibitor, we first compared the selected peptides with acarbose (IC50 = 0.672 mM, R^2^ = 0.98). The three peptides exhibited α-glucosidase inhibition but were not as effective as acarbose. Hence, they were compared with some identified promising peptides based on the IC50 ratio of peptide to acarbose to evaluate their potential as α-glucosidase inhibitors. The wheat germ peptides LDLQR, AGGFR, and LDNFR are often used as comparators in such studies using IC50 ratios of these peptides to acarbose (2.52, 2.54, and 2.70 respectively) [9]. The ratio values were close to that of Binglangjiang buffalo casein peptides (RNAVPITPTLNR, TKVIPYVRYL, YLGYLEQLLR, and FALPQYLK), which exhibit values of approximately 2.5 [43]. Compared with the above peptides, AAGRLPGY and RWPFFAFM in this study (their ratio was 1.96 and 1.93) showed excellent inhibitory potential. Another study reported that the ratios of three peptides VPKIPPP, LSMSFPPF, and MPGPPSD from *Ginkgo biloba* seed protein were 3.89, 1.22, and 2.54, respectively [29]. These findings suggest that AAGRLPGY and RWPFFAFM in this study represent potential enzyme inhibitors with excellent bioability. Moreover, VVRDFHNA with a ratio value of 3.16 was considerably more effective than VPKIPPP from *Ginkgo biloba* seed protein. Therefore, although the IC50 values of the screened peptides in this study were higher than that of acarbose, they can be considered potential α-glucosidase inhibitors, especially AAGRLPGY and RWPFFAFM.

The bioavailability of bioactive peptides in the human body are affected by chymotrypsin, pepsin, and trypsin and other exopeptidases after oral intake [44]. For example, trypsin can cleave the peptide to make it C-teriminal as arginine or lysine [45]. It seems that these three selected peptides are easily broken during the digestion process. Intestinal digestion may increase the activity of bioactive peptides by exposing more enzyme-binding sites [36,46]. In addition, some studies have proposed that peptides containing proline are more resistant to gastrointestinal digestion, which is consistent with the findings of AAGRLPGY and RWPFFAFM [47,48]. Therefore, these peptides need to be studied in animal models and in the clinical setting not only to determine their bioavailability in vivo but also to detect their toxicity and allergenicity. Peptide modification and replacement of drug delivery systems can be used to increase the bioavailability of peptides [44].

## 4. Conclusions

Through response surface method (RSM) optimization, the following alcalase hydrolysis conditions yielded the highest DH value of 26.5%: substrate protein concentration of 0.5%, enzymatic hydrolysis temperature of 53.0 °C, enzymatic hydrolysis time of 4.7 h, enzyme addition of 17,800 U/g, and pH of 10.5. The peptides less than 3 kDa obtained from hydrolysates showed the most promising inhibition. Then, the inhibitory peptides were screened by LC-MS/MS, molecular docking, and molecular dynamic simulation. In this study, AAGRLPGY (803.91 Da), VVRDFHNA (957.04 Da), and RWPFFAFM (1101.32 Da) were identified as potential α-glucosidase inhibitors, and AAGRLPGY and RWPFFAFM exhibited higher bioability in vitro. In the future, animal and clinical tests are needed to assess their bioavailability, toxicity, and allergenicity in the human body. These findings provide support for the development of functional foods from mulberry leaf protein.

## Figures and Tables

**Figure 1 foods-12-03917-f001:**
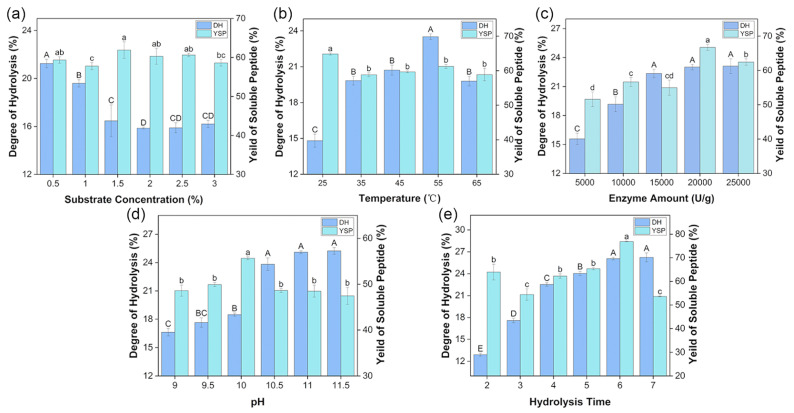
Single-factor experiment results. (**a**) Effect of substrate concentration on the degree of hydrolysis (DH) and yield of soluble peptides (YSP). (**b**) Effect of temperature on the DH and YSP. (**c**) Effect of enzyme amount on DH and YSP. (**d**) Effect of pH on the DH and YSP. (**e**) Effect of enzymatic time on the DH and YSP. Different letters indicate the significant difference among samples, *p* < 0.05. Capital letters are used for DH, and lowercase letters are used for YSP.

**Figure 2 foods-12-03917-f002:**
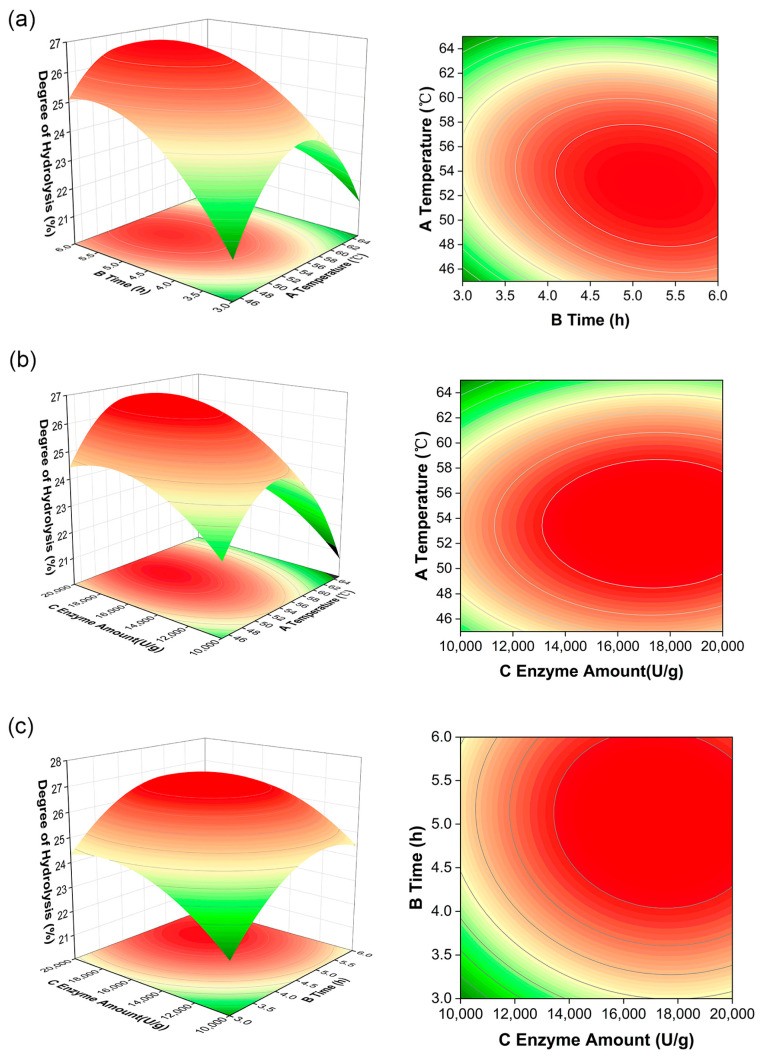
Results of the response surface experiment. (**a**) Response surface and contour plot of the effects of reaction time and enzymatic temperature on the DH. (**b**) Response surface and contour plot of the effects of reaction temperature and enzyme amount on the DH. (**c**) Response surface and contour plot of the effects of enzyme amount and reaction time on the DH. A stands for enzymatic hydrolysis temperature, B stands for enzymatic hydrolysis time, and C stands for enzyme amount.

**Figure 3 foods-12-03917-f003:**
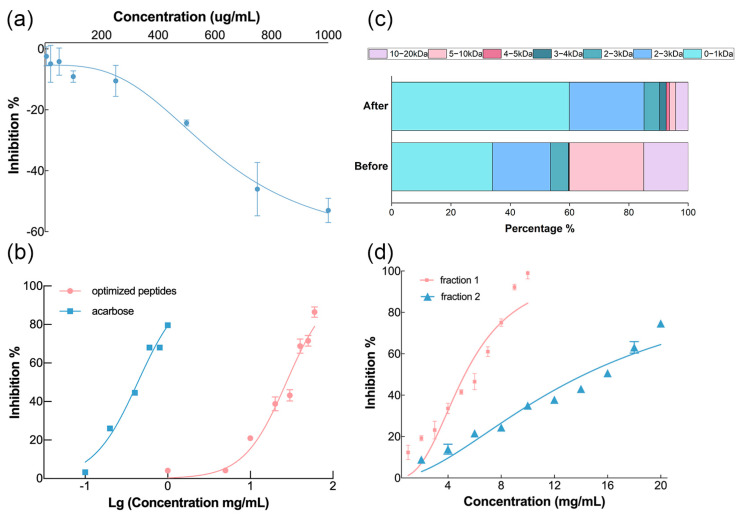
Results of the evaluation of enzymatic hydrolysis products before and after optimization. (**a**) α-Glucosidase inhibition rate curve of enzymatic hydrolysis products before response surface optimization. (**b**) α-Glucosidase inhibition rate curve of enzymatic hydrolysis products after response surface optimization and acarbose (IC50 of optimized peptides = 27.33 mg/mL, IC50 of acarbose = 0.43 mg/mL.) (**c**) The molecular weight distribution of enzymatic hydrolysis products before and after response surface optimization. (**d**) α-Glucosidase inhibition curves of different fractions (IC50 of fraction 1 = 5.27 mg/mL, IC50 of fraction 2 = 14.27 mg/mL).

**Figure 4 foods-12-03917-f004:**
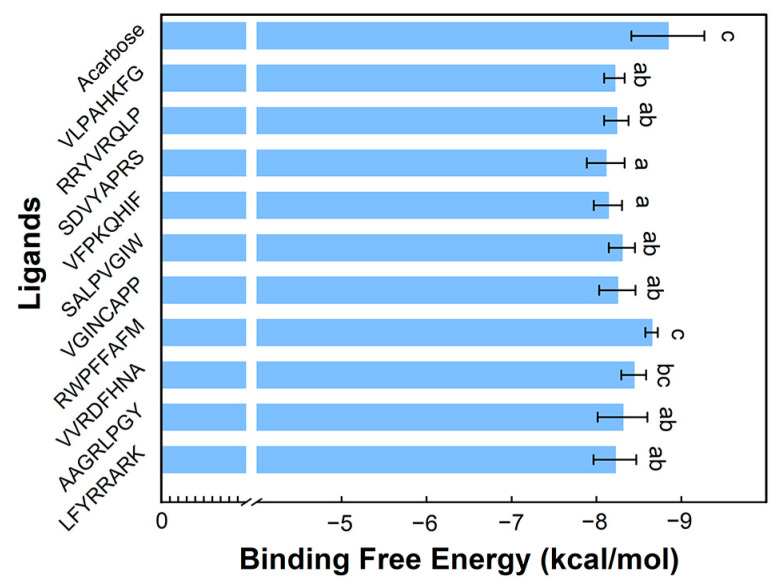
Results of evaluation of molecular docking binding free energy. Acarbose serves as the positive control. Different letters indicated the significant difference among samples, *p* < 0.05.

**Figure 5 foods-12-03917-f005:**
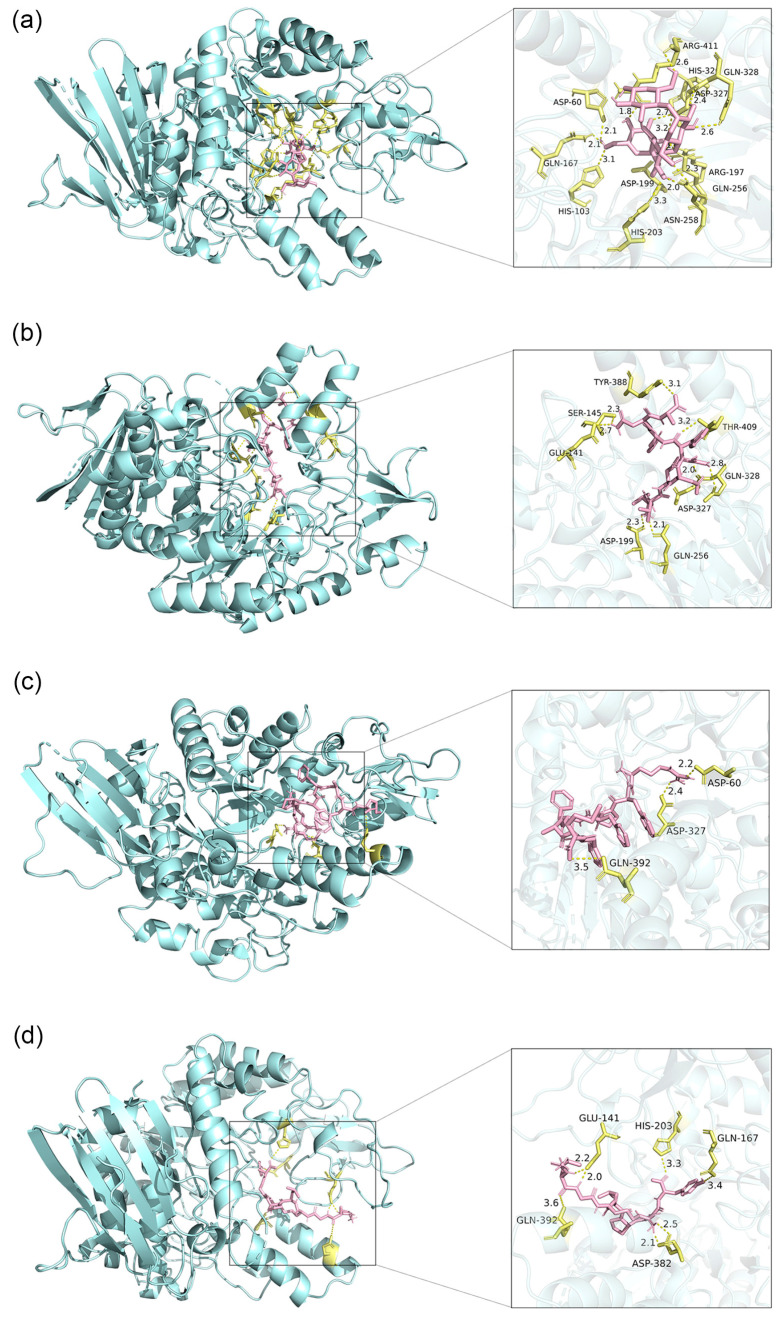
Docking results of the ligand-protein complex systems. (**a**) Ligand is acarbose. (**b**) Ligand is VVRDFHNA. (**c**) Ligand is RWPFFAFM. (**d**) Ligand is AAGRLPGY. Ligands are shown in pink, and residues of the enzyme are presented in yellow. The yellow dotted lines represent hydrogen bonds.

**Figure 6 foods-12-03917-f006:**
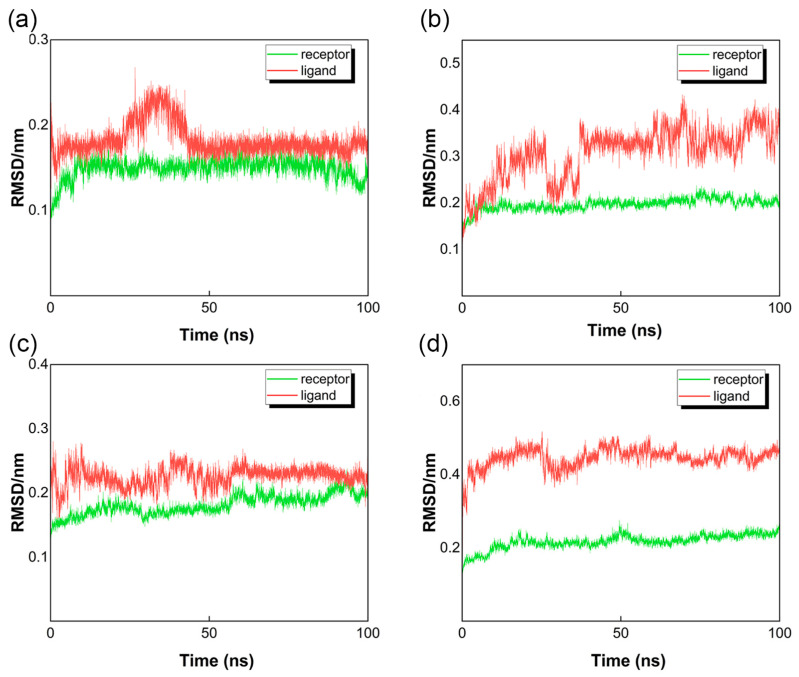
Root mean square deviation (RMSD) of ligand-protein systems. (**a**) Ligand is acarbose. (**b**) Ligand is AAGRLPGY. (**c**) Ligand is VVRDFHNA. (**d**) Ligand is RWPFFAFM.

**Figure 7 foods-12-03917-f007:**
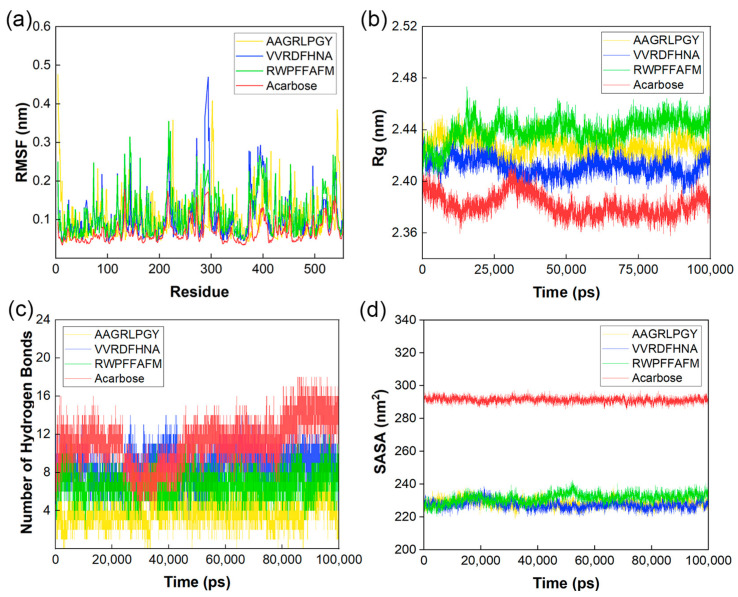
(**a**) Root mean square fluctuation (RMSF) of complexes. (**b**) Radius of gyration (Rg) of complexes. (**c**) The number of hydrogen bonds of complexes. (**d**) Solvent accessible surface area (SASA) of complexes.

**Figure 8 foods-12-03917-f008:**
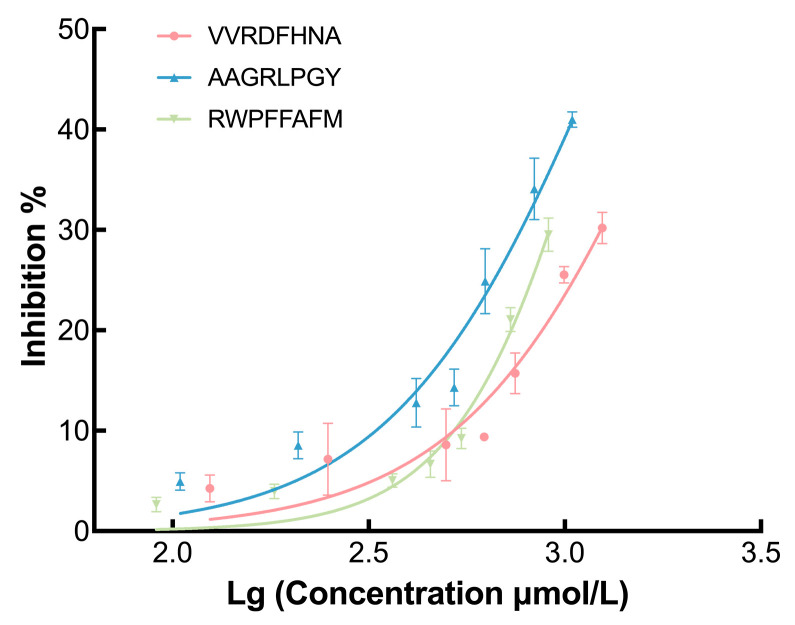
α-Glucosidase inhibition curves of selected peptides (IC50 of VVRDFHNA = 2.123 mM, IC50 of AAGRLPGY = 1.319 mM, and IC50 of RWPFFAFM = 1.299 mM.

**Table 1 foods-12-03917-t001:** The experimental design and outcomes of the response surface experiment.

Coefficient Source	Sum of Squares	Degrees of Freedom	Mean Square	F-Value	*p*
Models	76.62	9	8.51	78.44	<0.0001
A-Temperature	5.97	1	5.97	55.00	0.0001
B-Time	8.82	1	8.82	81.27	<0.0001
C-Enzyme Amount	7.94	1	7.94	73.16	<0.0001
AB	2.74	1	2.74	25.24	0.0015
AC	0.026	1	0.026	0.24	0.6420
BC	0.087	1	0.087	0.80	0.4003
A^2^	35.28	1	35.28	325.04	<0.0001
B^2^	7.25	1	7.25	66.78	<0.0001
C^2^	4.42	1	4.42	40.72	0.0004
Residual sum of squares	0.76	7	0.11		
Misfit term	0.46	3	0.15	2.08	0.2458
Pure error	0.30	4	0.074		
Total deviation	77.37	16			
	*R*^2^ = 0.9902			R^2^_adj_ = 0.9776	

## Data Availability

The datasets generated for this study are available on request from the corresponding author.

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
