# Peer review of "Development and Identification of Novel α-Glucosidase Inhibitory Peptides from Mulberry Leaves"

_foods, 2023, doi:10.3390/foods12213917_

Round 1

Reviewer 1 Report

The study has addressed in detail the potential use of mulberry leaf protein hydrolysates as inhibitors of starch metabolism in the small intestine. The work is novel and useful in the present prevailing diabetes challenges.

The titles for Figures 3 and 4 should not read 'evaluation of...', but 'results of evaluation of...'

Reviewer 2 Report

The authors extracted peptides from mulberry leaf protein and tested for α-glucosidase inhibition, followed by identification of the potentially effective peptide sequences using LC-MS/MS. The computational methods, docking and MID simulation, were used to show stable binding between the peptide and the α-glucosidase enzyme. After that, selected peptides were synthesized and determined for the inhibitory effects with the α-glucosidase using the same in vitro binding assay. Surprisingly, they found that the IC50 values of those peptides were higher than Acarbose (a positive control), which meant that the peptides bound less strongly than the acarbose. I think this was due to the different types of compound inhibitors, acarbose being a sugar molecule, whereas peptides exhibited different functional groups and may differ in the key interactions. 

Overall, the work is done appropriately. It's just that the authors could not find a new peptide that showed better inhibitory activity, unfortunately. 

My suggestion is to try finding some well-known standard peptides in the literature that showed the inhibition against α-glucosidase to additionally compare, not just acarbose.

Please perform some additional experiments, especially the inhibition assay with other known peptides to compare.

Reviewer 3 Report

The present authors prepared enzymatic hydrolysates of mulberry leaf protein. Their degree of hydrolysis was optimized by the response surface method. Consequently, they prepared some enzymatic hydrolysates with a high degree of hydrolysis. They found that low molecular weight fraction (<3 kD) of the hydrolysate showed in vitro  a-glucosidase inhibitory activity, while high molecular weight fractions activated a-glucosidase. They identified more than 60 peptides in the low molecular fraction. They did not use conventional activity-guided fractionation but use molecular docking method for prediction of a-glucosidase inhibitory activity. Some predicted peptides were confirmed to exert a-glucosidase inhibitory activity by wet experiment using synthetic peptides. This work apparently looks smart to identify bioactive peptides without using tedious and time-consuming methods. Recently similar studies have been increasing and similar data have been accumulated. However, this study has a big weak point. No consideration of stability of the peptides. Peptides are degraded not only by endo proteinases such as pepsin, trypsin, and chymotrypsin but by exopeptidases such as aminopeptidases, carboxypeptidases, and prolidases. Three peptides identified in this study have Arg in sequence, which is the cleavage site of trypsin. If the a-glucosidase inhibitory peptides in mulberry hydrolysate are degraded in gastrointestinal tracts before interaction to a-glucosidase, the a-glucosidase inhibition may not be observed in body. If so, the detailed in silico analyses performed in this study are not only meaningless but may also mislead the reader. Therefore, it should be shown that at least a low-molecular-weight active fraction can suppress the rise in postprandial blood sugar at a realistic dose. Experimentally, I don't think such animal study was difficult. If this is not possible, the authors should emphasize the possibility that peptides are frequently degraded, especially by exopeptidase, and strongly indicate the limitation of this study (lack of consideration on bioavailability).

Minor point. For fractionation of peptides by size exclusion chromatography using a column 7.8 mm x 300 mm, did you really inject 1 mL of sample? It is not usual.

Not so bad.

Reviewer 4 Report

The present work is very interesting and intended to investigate optimization of enzymatic hydrolysis process of mulberry leaf protein by alcalase  through the response surface method (RSM). The peptides less than 3 KDa from hydrolysates, AAGRLPGY, VVRDFHNA, and RWPFFAFM had been identified as potential α-glucosidase inhibitors.

Results obtained are well explained and data interpretation is also correct. Conclusions are consistent with the evidence and arguments presented. The methods used are sufficiently documented and allow replication studies.

About strenghts the authors explored the topic and they obtained the purpose of the study. About limitations the authors should explain the captions more comprehensively by clarifying in the graphs what each uppercase and lowercase letter refers to (Figures 1,2 and 4).

About reference 36 the date of their publication is too old and authors should modify it.

Round 2

Reviewer 2 Report

The revised version is satisfying for me now.

Reviewer 3 Report

The author has clearly stated the limitations of this research in response to my points regarding bioavailability, so I agree to accept them. On top of that, I would like to point out that you should respond logically to criticism, and that I don't think it's necessarily correct to use the reason given in already published papers as an excuse.

I'm very sorry for the late reply. The author has clearly stated the limitations of this research in response to my points, so I agree to accept it. However, there are questions about the meaning of publishing a paper with such limitations.